# Does mental fatigue affect skilled performance in athletes? A systematic review

He Sun[1]*, Kim Geok Soh[1]*, Samsilah Roslan[2], Mohd Rozilee Wazir Norjali Wazir[1], Kim Lam Soh[3]

1 Department of Sport Studies, Faculty of Education Studies, Universiti Putra Malaysia, Selangor, Malaysia, 2 Department of Foundation of Education, Faculty of Education Studies, Universiti Putra Malaysia, Selangor, Malaysia, 3 Department of Nursing, Faculty of Medicine and Health Science, Universiti Putra Malaysia, Selangor, Malaysia

* verson.upm@gmail.com (HS); kims@upm.edu.my (KGS)

## Abstract

### Background

Mental fatigue is a psychobiological state induced by a prolonged duration of demanding cognitive tasks. The effects of mental fatigue on physical performance have been well investigated in the literature. However, the effect of mental fatigue on skilled performance in sports remains unclear.

### Objective

This study aimed to report a comprehensive systematic review investigating the carryover effects of mental fatigue on skilled performance among athletes.

### Methods

A thorough search was conducted on PubMed, Web of Science, EBSCOhost (CENTRAL, SPORTDicus), and Scopus to select relevant literature, as well as on Google Scholar and sources of reference for grey literature. The selected literatures are centred on a mental fatigue protocol in which cognitive tasks are performed prior to athletic tasks. Only studies that used an experimental design to test two conditions, namely mental fatigue and non-mental fatigue, were selected.

### Results

Eleven articles were chosen based on the selection criteria. Mental fatigue affects skilled performance in three sports: soccer, basketball, and table tennis. A decline in skilled performance (decreased accuracy, increased performing time etc) is relevant to impaired executive functions. Seven studies focus on offensive skills, whereas only two studies are associated with defensive skills.

### Conclusion

Mental fatigue has a negative effect on various sports skills of high-level athletes, including their technical and decision-making skills; however, the impact is greater on offensive skills

**Data Availability Statement:** All relevant data are within the manuscript and its Supporting Information files.

**Funding:** The author(s) received no specific funding for this work.

**Competing interests:** The authors have declared that no competing interests exist.

than that of defensive skills in terms of the role of athletes. Impaired executive functions may be responsible for the negative effects of mental fatigue on skilled performance.

## Introduction

Skilled performance in sports is defined as the ability to perform at a high standard effectively and efficiently [1] and it determines the outcome in sport [2]. It is also known as skilled sports execution or technical performance [3]. For example, in ball sports such as basketball and rugby, the term specifically refers to how efficiently the athletes handle the ball, including executing effective disposals (i.e., passing, shooting, and dribbling) and tackling to the benefit of the team [4]. Most importantly, skilled performance requires quick and accurate decision making, which is the most important perceptual-cognitive skill in team sports [5, 6]. Moreover, Fernandez-Navarro et al. [7] found that some team athletes have different playing styles based on their roles (offensive and defensive). Similarly, a recent study highlighted that the required technical skills vary depending on their roles and a study that neglects the differences between the different roles and positions of athletes in a team may produce inconclusive findings and/or methodological problems [1]. Therefore, it may be insightful to investigate skilled performance in sports that include technical and decision-making skills related to their roles. Significantly, it has been well investigated that athletes' skills often decreases in the final stages of performance, with the decline being attributed to fatigue, as well as the athletes' decision-making skills [8, 9].

Fatigue is a complex and multifaceted phenomenon that is referred to as a decreased ability for maximal performance and the inability to complete a task that was once achievable within a reasonable time frame [10]. The influence of fatigue on sports performance has primarily been measured from both a metabolic and a neuromuscular standpoint (physical fatigue) [11]. However, recent discussions on the role of mental fatigue in sports performance have diverged [12, 13] from the common factor of "physical fatigue" [10].

Mental fatigue is a psychobiological condition caused by a prolonged duration of cognitive activity that is characterized by feelings of "tiredness" and "lack of energy" [14, 15]. Mental fatigue impairs cognitive performance, which has been linked to altered executive functions [16, 17], such as reduced directed attention and less accurate reactions. Since executive functions are highly correlated to a series of actions, such as the ability to initiate and stop, monitor and change behaviour, and plan for subsequent moves [18]. They are essential to performances in sports [19–21]. Researchers have recently begun to investigate the effect of mental fatigue on soccer performance."

Gonzaga [22] and Vestberg [23] explicitly suggested that executive functions are highly related to skilled performance. For example, in ball sports, new decisions must be made about passing the ball when the actions of opponents or the location of the ball tend to inhibit the intended actions (inhibition of executive functions). However, mental fatigue impairs inhibition [24]. To the best of the authors' knowledge, there is no comprehensive review of skilled performance in the context of mental fatigue.

Most studies have only focused on physical performance aspects, such as cycling time-to-exhaustion and resistance work [25–30]. In a previous investigation, skilled performance was analysed as part of physical performance [31], providing only a surface understanding of the issue and was insufficient to draw valid conclusions. In another comprehensive review, Van Cutsem et al. [29] evaluated 11 studies on physical performance conducted using experimental

design and found that while endurance performance was impaired, maximal strength and anaerobic work were not. However, the study did not include skilled performance.

The most recent study comes from Habay et al. [32], who conducted a comprehensive review of the effect of mental fatigue on sport-specific psychomotor performance. Habay and his colleague defined psychomotor performance as a highly complex motor behaviour that results from the cognitive processing of sensory and perceptual information in a sport-specific context. The results showed that mental fatigue impairs sport-specific psychomotor performance in many sports, and these impairments may be brain-related. However, the specific impact in brain was not discussed. Also, the study did not focus on skilled performance based on the role/position of players.

One prominent theory of mental fatigue in the physical field (endurance performance) is psychobiological model that was proposed by Marcora et al. [15]. The model postulates that endurance performance is a consciously regulated behaviour primarily determined by two psychological factors: motivation and perception of effort. Previous studies have found that mental fatigue dose not reduce both success and intrinsic motivation [15, 33]. Thus, a higher perception of effort is the most likely explanation for the impairment in the mental fatigue condition. However, the impairment in the skilled performance was theoretically attributed to a reduction in executive functions due to mental fatigue tasks activating the anterior cingulate cortex (ACC), likely leading to elevated adenosine and correspondingly decreased dopamine [34, 35]. To determine the mechanism of effects, the current study reviews each key executive functions in specific skilled performance.

## Methodology

The review's reporting follows the preferred reporting items checklist used in the systematic (PRISMA) protocol [36]. A systematic literature search was carried out involving four main databases, namely PubMed, Web of Science, EBSCOhost (CENTRAL, SPORTDicus), and Scopus for published works since their inception dates to January 2021 (S1 Table). EBSCOhost comprises numerous sub-databases. However, due to the content's relevance, only Cochrane Central Register of Controlled Trials (CENTRAL) and SPORTDicus were selected. Data searching was assisted by experienced librarians, who ensured the method's reliability.

### Eligibility criteria

The PICO method was used in the selection of literature (Table 1). The selected literature must also be available in English and fully published in a peer-reviewed journal. Specifically, studies were eligible when the study outcomes included any form of skilled performance (see Methodology: Data Analysis) in sports, which is the main outcome of the current review. Interventions were used to induce mental fatigue prior to the test of skilled performance with all types of cognitive tasks, and the population only included athletes who are very good at sports, particularly on the court. They can be classified into various levels, such as amateur, well-trained, elite and

**Table 1. PICO (Participation, Intervention, Comparison, Outcome).**

| PICO | Criteria |
|---|---|
| Participation | Athlete |
| Intervention | Acute fatigue induced by cognitive tasks |
| Comparison | Mentally fatigued vs non-mentally fatigued athletes |
| Outcome | Performance of sport skills |

professional etc. In addition, these studies had to include a control group that induced no or less mental fatigue than the mentally fatiguing tasks.

## Data analysis

To investigate a more in-depth investigation, the current qualitative review will examine the main outcomes of three aspects: offensive, defensive and decision-making skills (see details in the Introduction part). The action of engaging an opposing individual/team with the objective of scoring points or goals is referred to as offensive skill, whereas, defensive skill refers to the action of preventing an opponent from scoring [2]. In addition, athletes are usually required to pay sustained attention for making accurate, quick decisions based on the retrieval and processing of information from a dynamic environment [5, 6]. Athletes must excel in three aspects to achieve peak performance.

## Search strategy and selection of literature

The following keywords, truncation and Boolean operators, were used individually and in combination in the search in five databases (S1 Table). A search was also conducted on Google Scholar and based on the sources of reference in the selected reviews for additional literature that may not have appeared in the search results using the main databases.

The PICO method was used to determine whether the articles should be included or excluded (Table 1). First, the titles and abstracts were screened. Then, potential full-text articles were identified and selected. Two reviewers worked independently in this process (S.H. and S.K.G.). Disagreements were resolved by a discussion between the two reviewers. If necessary, a third reviewer (S.K.L.) was consulted to reach a consensus. Fig 1 summarises the selection process.

## Protocol and registration

The protocol of methods and planned analyses applied in this systematic review was registered in PROSPERO (ref. CRD42020201715). Although there are protocols in PROSPERO that examine the effects of mental fatigue, none of them focus on skill performance in sports. Hence, the novelty of the proposed protocol was ensured.

## Quality assessment

A systematic review is essential for evaluating relevant studies. Similarly, an overview should brief the assessment on the quality of the selected systematic reviews [37]. In addition, the data should be handled with care to prevent a "garbage in, garbage out" scenario that is the basis of conducting a systematic review [38].

The quality of the methodology used in this study was assessed using the quantitative assessment tool "QuaIlSyst" [39], which includes 14 items (Table 2). The scoring is based on how well a specific criterion was met (no = 0, partial = 1, yes = 2). The letter "NA" represents items that do not apply to the study design and are thus excluded from the calculation of the summary score. Each study's summary score was calculated by adding the total score obtained and dividing it by the total possible score. The scores, $\leq 55\%$, 55–75%, and $\geq 75$, indicate low, medium, and high quality, respectively. Any low-quality study should be excluded from the systematic review [39].

## Result

### Selection of literature

There were initially 1242 hits from PubMed, Web of Science, EBSCOhost (CENTRAL, SPORTDicus), and Scopus, as well as two hits from a reference search and Google Scholar. All

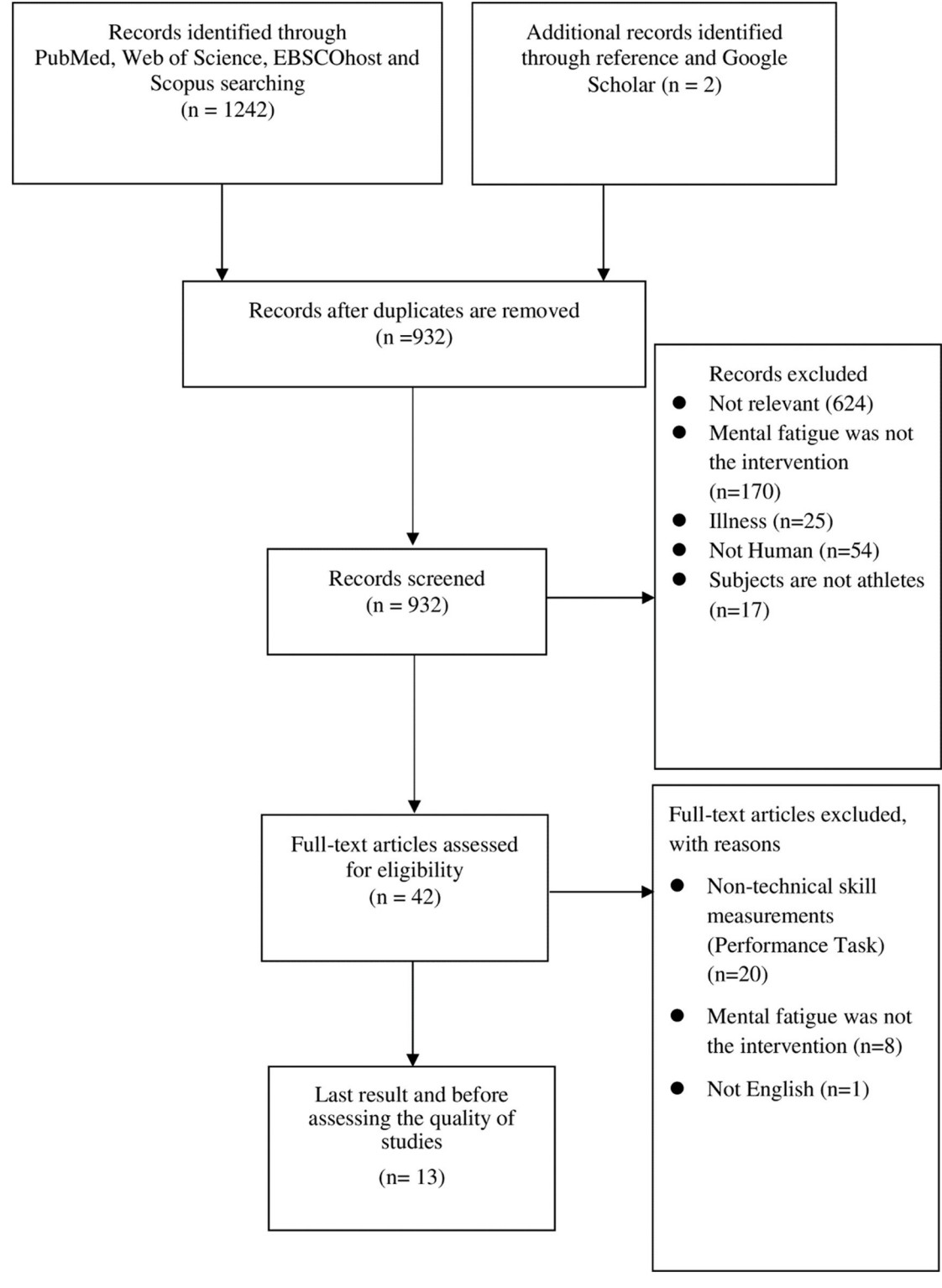

**Fig 1. PRISMA summary of the study selection process.**

duplicates were removed, and after screening the title and abstract, the entire text was read; ten literatures were chosen for this study (Fig 1). However, 2 out of the 13 literatures were deemed

**Table 2. "Qualsyst" of quality assessment.**

| Publication | Question/objective described | Appropriate study design | Appropriate subject selection | Characteristics sufficiently described | Random allocation | Researchers blinded | Subjects blinded | Outcome measures well defined and robust to bias | Appropriate sample size | Analytic methods well described | Estimate of variance reported | Controlled for confounding | Results reported in detail | Conclusion supported by results? | Rating |
|---|---|---|---|---|---|---|---|---|---|---|---|---|---|---|---|
| Badin et al. [40] | 1 | 2 | 2 | 1 | NA | 1 | 0 | 2 | 1 | 2 | 2 | 1 | 1 | 2 | medium |
| Smith et al. [41] | 2 | 2 | 2 | 2 | NA | 2 | 1 | 2 | 1 | 2 | 2 | 1 | 2 | 2 | high |
| Smith et al. [42] | 2 | 2 | 2 | 2 | NA | 0 | 1 | 2 | 1 | 2 | 2 | 1 | 2 | 2 | high |
| Smith et al. [43] | 2 | 2 | 2 | 1 | NA | 2 | 1 | 2 | 1 | 2 | 2 | 1 | 1 | 2 | medium |
| Le Mansec et al. [44] | 2 | 1 | 2 | 2 | NA | 0 | 0 | 2 | 1 | 2 | 2 | 0 | 2 | 2 | medium |
| Greco et al. [45] | 1 | 1 | 1 | 2 | 1 | 0 | 0 | 1 | 0 | 2 | 1 | 0 | 1 | 1 | low |
| Moreira et al. [46] | 2 | 2 | 2 | 2 | NA | 0 | 1 | 2 | 2 | 2 | 1 | 2 | 2 | 2 | high |
| Fortes et al. [47] | 2 | 2 | 2 | 2 | NA | 2 | 0 | 2 | 1 | 2 | 2 | 2 | 2 | 1 | high |
| Gantois et al. [48] | 2 | 2 | 2 | 2 | NA | 2 | 0 | 2 | 2 | 2 | 2 | 2 | 2 | 1 | high |
| Bahrami et al. [49] | 1 | 0 | 1 | 1 | 1 | 0 | 0 | 1 | 1 | 0 | 2 | 0 | 0 | 1 | low |
| Filipas et al. [50] | 2 | 2 | 2 | 2 | NA | 2 | 0 | 1 | 2 | 2 | 0 | 1 | 1 | 1 | medium |
| Fortes et al. [51] | 2 | 2 | 2 | 2 | NA | 2 | 0 | 2 | 2 | 2 | 2 | 2 | 2 | 1 | high |
| Trecroci et al. [52] | 2 | 2 | 2 | 2 | NA | 0 | 0 | 2 | 1 | 2 | 2 | 0 | 2 | 2 | medium |

NA: not applicable, 2 indicates yes, 1 indicates partial, 0 indicates no Quality; Quality score: $\geq$ 75% high, 55% –75% medium, $\leq$ 55% low.

to be of low quality and were thus excluded from this review. Thus, this systematic review was based on 11 studies that focused on skilled performance among athletes in various sports. The details are presented in Table 3.

## Overview of sports

The comprehensive search yielded studies involving ball sports, of which only Le mansec et al. [44] reported on an individual sport (table tennis), while the others reported on team sports (soccer and basketball). Soccer was the subject of 9 out of 10 studies on team sports [40–43, 47, 48, 50–52].

Interestingly, the majority of studies have focused on high-level athletes, including professionals [44, 47, 48, 51] and elites [40, 46]. In addition, most studies have focused more on offensive skills rather than defensive skills.

## Intervention inducing and the manifestation of mental fatigue

Badin et al. [40], Moreira et al. [46] and Gantois et al. [48] and Filipas et al. [50] used a 30-minute computerised Stroop task to induce mental fatigue. Gantois et al. [48] also investigated a 15-minute exposure to examine the dose-response of the Stroop task. Smith et al. [41], Smith et al. [42], and Smith et al [43]. have investigated the 30-minute exposure of the printed version Stroop task. In addition, Le Mansec et al. [44] required that participants complete a 90-minute AX-continuous performance test (AX-CPT). Fortes et al. [47] and Fortes et al. [51] have studied the effect of exposure to social network apps on smartphones for 15, 30, and 45 minutes, in addition to a 30-minute video gaming session, which is more realistic in today's world. In general, continuous exposure to cognitive tasks for $\geq 30$ minutes successfully induces mental fatigue when compared to two 15-minute interventions, as observed in studies involving smartphone exposure [47] and a computerised Stroop task [48].

To prove the condition of mental fatigue, seven studies have chosen subjective manifestation, all of which used the Visual Analogue Scale (VAS) [40, 41, 45–49] rated from "0" to "100", signifying "not exhausted at all" to "completely exhausted". However, the other five studies used VAS to report motivation and no significant difference in performance was reported between the mentally fatigued and control group in the given tasks in five of the studies [40, 41, 43, 44, 52]. Interestingly, greater motivation was observed in the mentally fatigued group than in the control group. [46]; the findings demonstrated that motivation is not diminished by mental fatigue. It is consistent with the psychobiological model of endurance performance, and effects of mental fatigue on skilled performance are not caused by motivation. However, only one study reported that mental fatigue likely increased RPE [40]. Most physiological variables that were assessed were general parameters, such as heart rate and blood lactate, which did not change in the presence of mental fatigue. Interestingly, Moreira et al. [46] found a higher concentration of salivary testosterone in mental fatigue athletes. The general physiological indicators, such as heart rate and blood lactate, did not show significant differences.

Cognitive performance was measured in four of the selected studies, which included 11 experiments, to evaluate the effects of mental fatigue [46–48, 51]. The results demonstrate that the 30-minute cognitive tasks result in a significant reduction in reaction time and performance accuracy among the athletes.

## Skilled performance outcome

As a prediction, the extracted literature has mainly focused on the skill demands based on the role or position of the athletes, and technical skills have been divided into two categories

**Table 3. Overview of publications details.**

| Publication | Subjects | N | Manipulation | Duration (min) | Measurement | Sport-skill Task | Type of Sport | Type of Skills | Outcome | |
|---|---|---|---|---|---|---|---|---|---|---|
| Badin et al. [40] | Elite | 20 | Stroop task computer version | 30 | VAS | Soccer: small-sided games | Team Lower Limb | Offensive | Passing Accuracy %↓ | MF↑ ME↑ MO ↔ RPE↑ HR↓ |
| | | | | | | | | Defensive | Tackle success↓ Tackle (n)↔ | |
| Smith et al. (Study 2) [42] | Well-trained | 14 | Stroop task paper version | 30 | BRUMS | LSPT | Team Lower Limb | Offensive | MF↑ ME↓ MO↔ RPE↔ HR↔ Penalty time↑ Performance time↑ Points per shot↓ Shot speed↓ | |
| Smith et al. [41] | Professional | 12 | Stroop task paper version | 30 | VAS | Soccer: decision making skill | Team Lower Limb | Decision Making | MF↑ ME↑ MO↑ Overall response accuracy↓ Response time↑ Visual search ↔ | |
| Smith et al. [43] | Well-trained | 14 | Stroop Task Paper Version | 30 | VAS | LSPT, LSST | Team Lower Limb | Offensive | MF↑ ME↑ MO↔ Error↓ Perfect passes↑ Time↔ Missed target↓ | |
| Le Mansec et al. [44] | Professional | 22 | AX-CPT | 90 | VAS | Table Tennis: forehand strokes | Individual Upper Limb | Offensive | MF↑ RPE↔ Ball speed↓ Accuracy↓ Faults number↑ Total score↓ | |
| Moreira et al. [46] | Elite | 48 | Stroop task computer version | 30 | Cognitive Performance | Basketball: small-sided game | Team Upper Limb | Offensive | RT↑ Ac↓ RPE↔ HR↔ Salivary parameters↑ alpha-amylase↑ Athletes' efficiency↔ Turnover number↑ | |
| Fortes et al. [47] exp. 1 | Professional | 20 | Smartphone | 15 | Cognitive Performance | Soccer: full-sided game | Team Lower Limb | Decision Making | RT↔ Ac↔ RPE↔ HRV↔ Hy↔ PDM↔ | |
| exp. 2 | | | | 30 | | | | | RT↑ Ac↓ RPE↔ HRV↓ Hy↔ PDM↓ | |
| exp. 3 | | | | 45 | | | | | | |
| Gantois et al. [48] exp. 1 | Professional | 20 | Stroop task computer version | 15 | Cognitive Performance | Soccer: full-sided game | Team Lower Limb | Decision Making | RT↔ Ac↔ RPE↔ HRV↔ Hy↔ Passing accuracy↔ | |
| exp. 2 | | | | 30 | | | | | RT↑ Ac↓ RPE↔ HRV↔ Hy↔ PDM↓ | |
| Filipas et al. [50] | Sub-elite | U14: 12 | Stroop task computer version | 30 | VAS | Soccer: LSPT, LSST | Team Lower Limb | Offensive | Penalty, Performance time↔ | MF↑ ME↑ |
| | | U16: 12 | | | | | | | Penalty, Performance time↔ | |
| | | U18: 12 | | | | | | | Penalty, Performance time↓ | |
| Fortes et al. [51] exp1 | Professional | 25 | Smartphone | 30 | Cognitive Performance | Soccer: full-sided game | Team Lower Limb | Decision Making | RT↑ Ac↓ RPE↔ Lactate Blood↔ Hy↔ PDM↓ | |
| Fortes et al. exp2 | | | Video games | 30 | | | | | | |
| Trecroci et al. [52] | sub-elite | 9 | Stroop task Smartphone app | 30 | VAS | Soccer: small-sided game | Team Lower Limb | Offensive | Negative passes↑ Pass; Shot accuracy↓ | MF↑ ME↑ MO↔ RPE↔ |
| | | | | | | | | Defensive | Negative, positive tackles; tackle (n); tackle success↔ | |
| | | | | | | | | Decision Making | Negative passes↑ Passing; Dribbling accuracy↓ | |

LSPT: Loughborough Soccer Passing Test; LSST: Loughborough Soccer Shooting Test. MF: mental fatigue; ME: mental effort; MO: motivation; RT: reaction time; Ac: accuracy; Hy: hydration. RPE: rating of perceived exertion; PDM: Passing performance decision making index; Higher↑; lower↓; no significant difference ↔ in the condition of mental fatigue compared with control condition; VAS: Visual Analogue Scale; BRUMS: Brunel Mood Scale.

(offensive and defensive skills). It is worth noting that previous studies focused more on offensive skills than defensive skills. The details are presented in the following sections.

## Technical skill

**Offensive skill.**  Seven of the selected studies have evaluated the offensive skills used in the Loughborough Soccer Shooting Test (LSST) and Loughborough Soccer Passing Test (LSPT) [42, 43, 50], as well as in small-sided soccer [40, 52] and basketball games [46], and the stroke performance test in table tennis [44]. In the LSST, athletes must perform a series of moves and score a goal. Mental fatigue affects athletes' accuracy and speed when shooting a goal [43]. In the LSPT, athletes must perform 16 passes against 4 gymnasium-standard benches as quickly as possible. For each mistake, they will be penalised with additional time (s). Higher penalty time (s), several errors (n) and missed target/missed passes (n) were observed in a study involving an experimental group (mentally fatigued athletes) and a control group [42]. There was no significant difference in time taken to complete the task between the two groups (Mean ± SD: MF = 47.8 ± 4.9; Control = 47.9 ± 4.1). However, the experimental group (mentally fatigued) had more errors (MF = 4.0 ± 2.8; Control = 2.5 ± 2.1) and fewer perfect passes (MF = 5.6 ± 1.4; Control = 6.6 ± 1.5) [45]. Filipas et al. [50] recently examined LSPT and LSST in three different age groups (U14, U16 and U18). Interestingly, only reductions in U18 were found (Penalty time: MF = 15.3 ± 4.7; Control = 8.0 ± 3.1; Performance time: MF = 67.2 ± 7.4; Control = 57.2 ± 6.7).

Because small-sided games (SSG) are a popular training method for developing multiple fitness, technical, and tactical components in team sport [53], Badin et al. [40] and Trecroci [52] conducted two separate experiments in SSG. A decline in accurate passes [Mean Difference (MD): 2.1% and 7% from two studies] and shots (MD = 36.9% in Trecroci [52]) were reported regarding offensive skills. Another investigation using SSG on athletes' basketball performance found that athletes committed a high number of errors (high number of total turnovers) [Effect Size (ES) = 0.71, 0.29; 1.12] [46]. Table tennis was also investigated. Le Mansec et al. [44] compared the effects of various types of fatigue on skilled performance. Mental fatigue affects the ball speed [2.2 ± 3.5% (-4.4 ± 0.1), p = 0.035, dz = 0.669] and accuracy [-6.7 ± 7.7% (-11.6, -1.8), p = 0.001, dz = 0.886], as well as increasing the chances of athletes committing faults [+5.5 ± 6.3% (+1.4, +9.4), p = 0.014, dz = 0.850] in specific protocols.

**Defensive skill.**  Only two studies in the selected literature dealt with defensive performance. Badin et al. [40] found that the percentage of tackling success is about 28% lower in mentally fatigued athletes than that of athletes in controlled conditions, as observed in the soccer SSGs. In Trecroci et al. [52] the success rate was not significantly different between the two conditions (MF = 46.6 ± 27.1; Control = 49.1 ± 34.4). Moreover, neither study found any changes in the total number of tackles in terms of defensive skills.

**Decision-making skills.**  The first investigation on decision-making skills was conducted in a controlled laboratory setting [41]. Film simulation was used by the researchers, which is one of the most popular methods. The results demonstrated that athletes' decision-making ability is impaired after performing a cognitive task in terms of response accuracy (MF = 80.9 ± 6.4%; Control = 85.7 ± 4.9%; ES = -0.89 ± 0.73) and time (MF = 768 ± 134 ms; Control = 685 ± 156 ms; ES = 0.49 ± 0.47). Mental fatigue causes decrements because it impairs executive function, shifting athletes' goal-directed attention to stimulus-driven attention, negatively impacting decision-making skills [41].

Three of the selected studies have attempted to examine the effects of mental fatigue on decision-making skills in soccer athletes, particularly when passing the ball. Despite testing various treatments, the studies used the same method for measuring the outcome; two of the

studies have investigated the effects of smartphones and video games [47, 51] whereas the other looked at the effects of normal/experimental cognitive tasks on sports performance [48]. All three studies have found that mental fatigue influences passing decision making (condition effect: $p < 0.01$) in soccer athletes after $\geq$30 minutes of treatment exposure.

Based on the study of Smith et al. [41], Trecroci et al. [52] conducted an experiment in a SSG (4 vs. 4; 1 wildcard) to examine decision-making skills, which are more relevant to real competition. Mental fatigue affects not only the total number of passing decision making (MF = 2.4 ± 1.1; Control = 1.5 ± 1.2), but also the accuracy of passing (MF = 81.5 ± 9.5; Control = 88.2 ± 10.4) and dribbling (MF = 41.3 ± 32.2; Control = 75.3 ± 35.5).

## Discussion

This systematic review attempts to outline current knowledge of the effects of mental fatigue on skilled performance in athletes, including technical and decision-making skills.

### Overview of sports

Unsurprisingly, all of the selected 11 studies have focused primarily on ball sports (soccer, basketball and table tennis), as the cognitive demands in ball sports are extremely challenging [54, 55]. In sports, the trajectory paths of athletes' movements and the ball are usually unpredictable; interruptions may occur, causing changes in direction, as well as various occlusions and segmentation, such as objects blocking or disappearing from view. Therefore, athletes playing ball sports must remain alert throughout the game [34, 54] that increases the chances of mental fatigue. In line with previous studies, the predominant focus on soccer athletes demonstrates that these athletes may be more susceptible to mental fatigue than other athletes due to the longer duration of exposure to cognitive tasks in ball sports than other sports [34].

Most of the selected studies have focused on high-level athletes, which suggests that the chances of mental fatigue are exceptionally high in athletes who are under extreme pressure due to high-intensity training and competition demands [13, 34]. However, studies on athletes in other categories are also important, because many studies have shown that factors unrelated to sports activities are also responsible for mental fatigue in athletes during games [47, 51, 56, 57]. Future studies should consider this particular scope.

### Intervention inducing and the manifestation of mental fatigue

The Stroop task was used in most of the selected studies (see Table 3). Generally, the Stroop task is a classical method for measuring inhibitory control and sustained attention, which are executive functions [18]. According to a previous study, these functions are highly related to skilled performance in sports [22, 23, 34]; a decrease in such abilities may impair athletes' performance. Also, in line with recent studies, a 30-minute Stroop task can be used to induce effective mental fatigue in athletes of all levels [29]. Contrastingly, McMorris [35] indicated that shorter cognitive tasks have a larger negative effect than longer ones due to learning effects. The longer duration influence was found to be more negative in the current review [47, 48]. Because the types of cognitive tasks and subsequent tests are different. Brown et al. [31] found that shorter and longer cognitive tasks did not affect submaximal exercise. McMorris [35] also raised this issue, stating that the short versus long duration of cognitive tasks should be addressed in the future.

Consistent with Smith et al. [58], the current review showed that subjective manifestation is reliable for measuring mental fatigue. Previous studies have shown that subjective ratings can detect fatigue when objective assessment is not possible [59, 60], Moreover, it has been suggested that subjective mental fatigue precedes any decrements in performance [60].

Particularly, VAS appeared to be an effective instrument, and it has been widely implemented [40, 41, 43, 44, 50, 52], as well as for minimum time spent and easy interpretation [59].

## The effect of mental fatigue on sports skilled performance

**Technical skill.** Technical skills are mainly divided into offence and defence skills based on specific demands based on the athletes' role or position. Notably, the majority of the selected studies have focused on offensive skills because cognitive resources are more exhausted when performing offensive skills than defensive skills [61]. Therefore, athletes are more likely to experience mental fatigue when performing offensive skills than defensive skills, because the perception of effort increases more in the former than the latter. However, this hypothesis should be confirmed in future studies by a direct comparison of these two technical skills used in the same sport.

Because mental fatigue impairs the trade-off between speed and accuracy [62]. More complex sport-related technical skills will almost certainly be altered. Le Mansec et al. [44], Smith et al. [43] and Filipas et al. [50] have all found that mental fatigue reduces accuracy and ball speed. However, Smith et al. [43] conducted a time-based experiment that emphasized speed. Generally, a speed-accuracy trade-off occurs during performance under mental fatigue conditions, whereby the speed performance (primary) goal is maintained at the expense of the accuracy (secondary) goal. This systematic review highlights the negative effects of mental fatigue on skilled performance in sports, in general. However, the effects vary depending on the task's demand. The information can be used to develop a training plan and a match. Furthermore, the knowledge enables athletes to adopt a good strategy in the event of mental fatigue; for example, during ball games, passing the ball to a team athlete in a less pressured position (more time, but less accurate), rather than risking a pass to a team athlete at a highly pressured attacking position (more accurate, but less time).

Mental fatigue does not affect physical performance in simulated games, as reported by Badin et al. [40] and Gantois et al. [48]. However, the effects can be observed in soccer-specific endurance tasks [42]. This is also in contrast to Trecroci et al. [52], who reported that athletes likely covered less distance when accelerating, and possibly a lower equivalent distance than in control conditions in the SSG. However, such effects are not elaborated on this systematic review, because they are beyond the research scope. However, future research should not disregard such findings to better understand the effects of mental fatigue.

**Decision-making skill.** Smith et al. [41] conducted the first study on decision-making performance in sports, demonstrating that mental fatigue negatively influences the speed and accuracy of athletes in making critical decisions in a lab setting. Trecroci et al. [52] extended their results to a field setting (SSG), which is more relevant to real-world competition. Poor performance was observed in both studies, probably due to impairments in cognitive skills such as executive functions and information processing [63, 64], which may influence athletes' decision-making skills [41]. Athletes' visual search behaviour has also been linked to lower information processing performance, according to researchers. In a previous study, although athletes in both the experimental (mental fatigue) and control groups extracted information from similar sources (athletes with the ball, opponents, the ball), the ability to identify and to use the information appropriately was compromised by the athletes in the experimental group. The findings suggest that mental fatigue may affect attentiveness [41], which is controlled by the executive function of "inhibition".

Three of the selected studies have focused on passing decision-making skills, despite the slight differences in their purposes. Two of the three studies sought to investigate prolonged exposure to smartphones and video games as a possible cause of mental fatigue [47, 51], as well

as the subsequent impact on decision-making skills; the third, a study by Gantois et al. [48], sought to directly analyse the effects of mental fatigue on decision making. These three studies have solely focused on decision-making ability related to ball passing because it is one of the key tasks that determines game outcomes [65]. All three studies have applied similar equations to calculate the passing decision-making index (PDM). The equation is as follows:

$$\mathrm{PDM} = \frac{A_a}{Aa + I_a} \times 100$$

Aa = appropriate actions
Ia = inappropriate actions

The findings of the three studies demonstrate that mental fatigue impairs ball passing decision-making skills when exposed to a 30-minute or 45-minute cognitive task; exposure to a 15-minute cognitive task has no effect on decision-making skills. Therefore, it is concluded that the duration of a cognitive task should be at least 30 minutes to experimentally induce mental fatigue.

## Potential mechanism

Understanding the potential underlying mechanisms may assist in the development of interventions to counteract the negative effect of mental fatigue on skilled performance. To consider the mechanisms, it is inevitable to consider the perception of effort in the literature of physical performance, which has demonstrated that it is the only parameter to mediate the effect of mental fatigue on endurance performance [15, 29]. However, the current review did not find consistent evidence, as the level of RPE was not significantly different in the majority of the studies. Importantly, Moreira et al. [46] found that mental fatigue elevates salivary testosterone that may be associated with the increased errors of technical skill. Indeed, testosterone is thought to be involved in regulating activity of the mesolimbic reward system involving dopamine [66]. Moreira et al. [46] proposed that lower testosterone levels may influence dopamine transmission in mentally fatigued athletes. This finding shows that mental fatigue decreases dopamine in the ACC, impairing executive functions.

Technical skill impairment has been strongly related to attentional direction [40, 42, 43]. Indeed, mental fatigue has been shown to shift attention from goal-directed to stimuli-directed that are irrelevent to subsequent performance [16, 67]. Furthermore, it decreased the ability to anticipate ball movement and prepare to control it [43]. Smith et al. [42] even suggested that this negative effect may be more likely in the match setting, where the additional irrelevant stimuli are increased compared to a control experimental environment. Moreover, athletes must constantly monitor and adjust their skills to improve their performance. However, a previous study indicated that mental fatigue leads to an impaired capacity for performance monitoring and adjustment [68]. The ability to monitor error performance is thought to rely predominantly on intact ACC functioning [69]. Lorist et al. [68] found that people typically slow down to perform when an error occurs. Thus, when athletes are mentally fatigued, they may slow down to exert technical skills to avoid errors, allowing opponents to counterattack.

Athletes' decision-making ability is also impaired due to the shift in attention focus. In soccer, decision making highly relies on the human brain's ability to perceive relevant information from a complex environment while blocking out irrelevant distractions [48, 70]. This process is controlled by selective attention [71, 72]. Selectivity prevents athletes from reacting reflexively to environmental stimuli and allows them to make flexible decisions. However, it was impaired by mental fatigue.

Another possible explanation is that athletes are unable to respond to cues to prepare for upcoming decisions because mental fatigue negatively impairs preparation and planning [73]. To make a good decision, athletes should interpret and/or anticipate the movement of teammates, opponents and/or the ball etc. However, the execution of preparation and planning processes is susceptible to mental fatigue and frontal negativity [73].

Furthermore, the impairment has been attributed to the reduced ability of the visual search strategy. In team sports, mentally fatigued athletes tend to spend more time fixating on their opponents while spending less time fixating on their teammates [41, 47]. Therefore, athletes have little focus on the movement of their teammates, even though they may understand the movements of their opponents, impairing the passing decision-making skill, as evidenced by three extracted studies [47, 48, 51].

## The implications and future directions

The mechanisms of mental fatigue in sports in terms of skilled performance are complicated. Understanding these mechanisms may assist in the development of future interventions. Prior cognitive tasks have been shown to activate the ACC, inducing the concentration of adenosine to rise. Indeed, the ACC plays a key role in many executive functions that are in charge of performing skills among athletes. Future studies may look into strategies, such as implementation intentions to aid in action automation; mindfulness music, and nature exposure to enhance directed attention; biofeedback to monitor/adjust performance and improve executive functions while countering mental fatigue.

The impact of duration after cognitive tasks should be considered by future studies, for the condition of metal fatigue is acute. Although the precise duration of mental fatigue is unknown, Smith et al. [58] reported the prolonged mental fatigue condition in an individual after 60 minutes in an assessment using the Psychomotor Vigilance task and VAS; the findings also demonstrate that the individual may recover from acute mental fatigue in 60–135 minutes.

The fact that most of the investigations in the selected literatures have converged on soccer suggests that soccer athletes may be more prone to mental fatigue than athletes in other ball sports, or athletes in other sports in general. Previous studies have consistently reported that soccer athletes face extremely challenging cognitive demands when competing [13, 34]. However, attention should also be given to athletes from other sports, as recent studies have shown that mental fatigue may be caused by factors unrelated to sports activities (i.e., smartphones, video playing, fixture congestion, etc.).

Most importantly, athletes who play an offensive role may be more susceptible to mental fatigue than defensive athletes in some team sports, such as soccer. This suggestion should be considered by coaches. Offensive athletes may need to do more mental training.

However, during high demand cognitive tasks, athletes are prompted to perform an executive function to maintain vigilance. If this process is prolonged, mental fatigue may set in. This knowledge provides opportunities for future studies to improve skill performance in sports by increasing athletes' ability to perform an executive function.

## Conclusion

This systematic review demonstrates that mental fatigue affects a wide range of components of skilled performance in sports among high-level athletes. Technical and decision-making skills are important components, particularly when athletes play offensive roles. The influences have been predominantly observed in three sports: soccer, basketball, and table tennis.

A 30-minute Stroop task is sufficient to induce mental fatigue in all cognitive assessments. In addition, VAS is effective for subjective measurement. The findings suggest that athletes should avoid cognitive tasks prior to competition or training, to achieve optimal performance. This includes exposure to screens and video games.

There were no physiological or psychological indicator mediated effects of mental fatigue on skilled performance. The possible mechanism for the effect of mental fatigue on skilled performance is the increased concentration of adenosine in the prefrontal cortex activated by cognitive tasks, impairing a series of executive functions. Interventions to counteract this negative effect of mental fatigue are urgently needed.

## Limitations

This systematic review poses a few noteworthy limitations. First, this systematic review, conducted rigorously, is not a meta-analysis. Moreover, the systematic review focuses only on skilled performance, defined as technical and decision-making skills, excluding physical performance in sports. Because three of these skilled performances are closely related and contribute to athletes' performance, future studies should investigate them together to obtain more comprehensive results. Finally, selecting articles only written in English may limit the representation of the results even further.

## Supporting information

**S1 Table. Detailed search strategy.**
(DOC)

**S2 Table. PRISMA 2009 checklist.**
(DOC)

## Acknowledgments

The authors would like to thank Dr Zubaidah Ibrahim and Dr Lim Xin Jean for their assistance with the searching strategy.

## Author Contributions

**Methodology:** Kim Geok Soh, Mohd Rozilee Wazir Norjali Wazir, Kim Lam Soh.

**Supervision:** Kim Geok Soh.

**Writing – original draft:** He Sun.

**Writing – review & editing:** Kim Geok Soh, Samsilah Roslan, Kim Lam Soh.

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
