## [Decision Letter · Decision Letter 0]

24 May 2021

PONE-D-21-04367

Does Mental Fatigue Affect Skilled Performance in Sports among Athletes? A Systematic Review

PLOS ONE

Dear Dr. Sun,

Thank you for submitting your manuscript to PLOS ONE. After careful consideration, we feel that it has merit but does not fully meet PLOS ONE’s publication criteria as it currently stands. Therefore, we invite you to submit a revised version of the manuscript that addresses the points raised during the review process.

We look forward to receiving your revised manuscript.

Kind regards,

Daniel Boullosa

Academic Editor

PLOS ONE

Journal Requirements:

3. Please include a copy of Table 4 which you refer to in your text on pages 10, 15 and 16.

Reviewers' comments:

Reviewer's Responses to Questions

**Comments to the Author**

1. Is the manuscript technically sound, and do the data support the conclusions?

Reviewer #1: Partly

Reviewer #2: Partly

2. Has the statistical analysis been performed appropriately and rigorously? 

Reviewer #1: N/A

Reviewer #2: N/A

3. Have the authors made all data underlying the findings in their manuscript fully available?

Reviewer #1: Yes

Reviewer #2: Yes

4. Is the manuscript presented in an intelligible fashion and written in standard English?

Reviewer #1: No

Reviewer #2: Yes

5. Review Comments to the Author

Reviewer #1: General comments

This is an interesting review study with results that might add to our current understanding of mental fatigue and skilled performance in athletes. There are however a number of issues that need attention.

A thorough revision of grammar and language of this manuscript is imperative before making further decisions on the acceptability of the manuscript. So please have the manuscript corrected by a native English speaker/a professional company for revising scientific documents.

Specific comments

Title

I suggest change to “Does mental fatigue affect skilled performance in athletes? A systematic review”

Abstract

Background

Last sentence. Change “mechanism” for “effect”.

Conclusion

“Conclusion” could be dark and lined up to the left.

Introduction

First paragraph

Change “ball sports” to “team sports”.

Change “Fernandez- Navarro et al7”to “Fernandez-Navarro et al. [7]”

At the end of the first paragraph recommends for describe that fatigue can impair technical and decision-making skill. This information is importante because could link the first with second paragraph.

Third paragraph

Very long sentence “Since these executive functions are highly correlated to a series of actions, for example the capacity of initiating and stopping, monitoring and changing behaviour, and planning for subsequent moves [15], which is the key to a good sports performance among athletes [16-18], researchers recently started to investigate the effect of mental fatigue on soccer performance.” It’s recommended to divide it without the information being altered.

Forth paragraph

Change “ball game” to “team sport”

Before sentence “To the authors’ knowledge, a comprehensive review on skilled performance in the context of mental fatigue is still unavailable.” the authors could describe the possible motives of mental fatigue impair the inhibitory control and consequently perceptual-cognitive skills (e.g., increased adenosina in ACC).

Seventh paragraph

Change “on performance in endurance performance” to “on endurance performance”

Change “ball sports” to “team sports”.

Although the most adopted framework model in the literature of mental fatigue is psychobiological, this model does not seem to explain the changes in perceptual-cognitive skills (e.g., decision-making). Perhaps the authors could explain the brain mechanisms pointed by Terry McMorris (e.g., attenuated dopamine to front-parietal regions) and visual search strategy (visual cortex) changed in athletes mentally fatigued.

References

McMorris, T. Cognitive Fatigue Effects on Physical Performance: The Role of Interoception. Sports Medicine, 2020. https://doi.org/10.1007/s40279-020-01320-w

Mitchell R. Smith, Linus Zeuwts, Matthieu Lenoir, Nathalie Hens, Laura M. S. De Jong & Aaron J. Coutts (2016): Mental fatigue impairs soccer-specific decision-making skill, Journal of Sports Sciences, DOI: 10.1080/02640414.2016.1156241

Methods

It’s recommended add “data analysis” for describe the subgroups created in results section:

Technical skill, offensive skill, defensive skill, and decision-making skill.

Results

Overview of Sports

First paragraph

Change “ball sports” to “team sports”

Intervention inducing and manifestation of mental fatigue

It’s suggested to create a Table indicating the main findings (e.g., task used to induce mental fatigue, test or scale utilized for measurement mental fatigue)

Discussion

Intervention inducing and manifestation of mental fatigue

Second paragraph

“However, contrast to the study of [50], the higher perception of effort is not only responsible for the impairment in physical performance, but also skilled performance. Four selected literature have measured the effects of motivation as part of the methodology [36, 40, 42, 44].”

The sentences are unconnected. It’s recommended re-writing.

I understanding that the impaired perceptual-cognitive skill in athletes mentally fatigue can be explain by visual search strategy altered or by activity changed in front-parietals regions (e.g., increased theta wave in PFC). It’s suggested describe about this.

The implication and future directions

First paragraph

It’s recommend to remove the first paragraph, because the findings of systematic review no demonstrated rating perceived exertion as a mediator between mental fatigue and skilled performance.

Second paragraph

There is a difference between rating perceived exertion (RPE) and session-RPE. For evaluate session-RPE is necessary measure RPE 30-min after physical or experimental session. Some studies measured RPE after 30-min for analyze session-RPE or internal training load. It’s suggested review this paragraph.

Reviewer #2: General Comments

This is a timely systematic review of the effects of mental fatigue on skilled performance in sport which I believe can have a good impact in the field. There are, however, several issues to be resolved before publication as explained in the specific comments. I also suggest the authors to have the whole manuscript reviewed by an English copyeditor as some terminology and phrases are often unusual and odd. Some parts are also unnecessary and can be eliminated (see specific comments).

Specific comments

INTRODUCTION

Although not entirely similar, the authors should include in their introduction and discussion reference to the literature review by Habay and colleagues 1 dealing with the effects of mental fatigue on psychomotor performance, including tests of skilled sport performance.

The paragraph on the single narrative review on sport-related performance can be eliminated or shortened

The psychobiological model proposed by Marcora is a model of endurance performance and cannot be transferred so easily to skilled sport performance. Although you can mention the (generally lack of) effect of mental fatigue on motivation, the most relevant explanations for the negative effects of mental fatigue on skilled sport performance are its effects on attention, inhibitory control, and other executive functions. In the context of skilled sport tasks, RPE is more an index of how much effort the athletes exerted during the tasks (which can be similar but with different performance outcomes due to mental fatigue) rather than a mechanism of impaired performance. So I suggest reducing the sections about the psychobiological model of endurance performance.

METHODS

One of the main methodological problems of this paper is the lack of proper operational definitions of the PICO criteria. Please provide in the revised manuscript. There should also be operational definitions for the categories you have used to analyse the results of the systematic review, e.g. level of the athletes, difference between offensive and defensive skills, etc. Fnally, you should also include information about the dates and language limits you used for the search.

RESULTS

In general, in the text I would like to see more quantitative information about the literature, for example absolute values and/or percentages of papers in relation to the analyses shown in Table 3.

DISCUSSION

Try not to repeat too much the results in the Discussion. Instead summarise and then compare the observed effects of mental fatigue with the effects of other related factors, e.g. i) the decline in technical performance between the first and second half of a soccer match, ii) the effects of sleep deprivation or iii) the effects of physical fatigue.

I also suggest that, for each skilled sport performance task, you identify some key executive functions and discuss literature from cognitive science on the effects of mental fatigue on these executive functions. Such discussion would provide a much better explanation of potential mechanisms than the current explanation based on the psycholobiological model which needs to be reduced in length and de-emphasised as the main explanation for the negative effects of mental fatigue on skilled sport performance (see also comments about the Introduction).

References

1. Habay, J. et al. Mental Fatigue and Sport-Specific Psychomotor Performance: A Systematic Review. Sports Med. (2021) doi:10.1007/s40279-021-01429-6.

6. PLOS authors have the option to publish the peer review history of their article (what does this mean?). If published, this will include your full peer review and any attached files.

Reviewer #1: **Yes: **Leonardo de Sousa Fortes

Reviewer #2: No

---

## [Author Response · Author response to Decision Letter 0]

2 Jul 2021

Thanks for giving valuable feedback regarding the manuscript. We greatly appreciate it. We have been able to incorporate changes to reflect most of the suggestions. After thorough considerations, we respond to each point including reviewer and editor comments in the attachment "Response to reviewers". Please have a look.

---

## [Decision Letter · Decision Letter 1]

24 Sep 2021

Does mental fatigue affect skilled performance in athletes? A systematic review

PONE-D-21-04367R1

Dear Dr. Sun,

We’re pleased to inform you that your manuscript has been judged scientifically suitable for publication and will be formally accepted for publication once it meets all outstanding technical requirements.

Kind regards,

Daniel Boullosa

Academic Editor

PLOS ONE

Additional Editor Comments (optional):

Reviewers' comments:

Reviewer's Responses to Questions

**Comments to the Author**

1. If the authors have adequately addressed your comments raised in a previous round of review and you feel that this manuscript is now acceptable for publication, you may indicate that here to bypass the “Comments to the Author” section, enter your conflict of interest statement in the “Confidential to Editor” section, and submit your "Accept" recommendation.

Reviewer #1: All comments have been addressed

2. Is the manuscript technically sound, and do the data support the conclusions?

Reviewer #1: Yes

3. Has the statistical analysis been performed appropriately and rigorously? 

Reviewer #1: Yes

4. Have the authors made all data underlying the findings in their manuscript fully available?

Reviewer #1: Yes

5. Is the manuscript presented in an intelligible fashion and written in standard English?

Reviewer #1: Yes

6. Review Comments to the Author

Reviewer #1: The manuscript improved significantly. The authors answered all suggestions. The paper can be accepted. Congratulations for authors.

7. PLOS authors have the option to publish the peer review history of their article (what does this mean?). If published, this will include your full peer review and any attached files.

Reviewer #1: **Yes: **Leonardo de Sousa Fortes

---

## [Editor Report · Acceptance letter]

29 Sep 2021

PONE-D-21-04367R1 

Does mental fatigue affect skilled performance in athletes? A systematic review 

Dear Dr. Sun:

I'm pleased to inform you that your manuscript has been deemed suitable for publication in PLOS ONE. Congratulations! Your manuscript is now with our production department. 

Kind regards, 

on behalf of

Dr. Daniel Boullosa 

Academic Editor

PLOS ONE